# Investigating finite-size effects in random matrices by counting resonances

**Anton Kutlin**[1][⋆] **and Carlo Vanoni**[2,3][†]

**1** Abdus Salam International Center for Theoretical Physics, Strada Costiera 11, 34151 Trieste, Italy

**2** SISSA – International School for Advanced Studies, via Bonomea 265, 34136, Trieste, Italy

**3** INFN Sezione di Trieste, Via Valerio 2, 34127 Trieste, Italy

⋆ anton.kutlin@gmail.com ,    † cvanoni@sissa.it

## Abstract

Resonance counting is an intuitive and widely used tool in Random Matrix Theory and Anderson Localization. Its undoubted advantage is its simplicity: in principle, it is easily applicable to any random matrix ensemble. On the downside, the notion of resonance is ill-defined, and the 'number of resonances' does not have a direct mapping to any commonly used physical observable like the participation entropy, the fractal dimensions, or the gap ratios (r-parameter), restricting the method's predictive power to the thermodynamic limit only where it can be used for locating the Anderson localization transition. In this work, we reevaluate the notion of resonances and relate it to measurable quantities, building a foundation for the future application of the method to finite-size systems. To access the HTML version of the paper & discuss it with the authors, visit https://enabla.com/pub/558.

## 1   Introduction

Whether a quantum system thermalizes or not under its own unitary dynamics is an issue that has received a lot of attention in the last decades and a general theory allowing to discern thermal from non-ergodic systems is still lacking [1, 2]. Thermal quantum systems are expected to satisfy the Eigenstate Thermalization Hypothesis (ETH) [3], stating that the expectation values of observables will evolve in time and ultimately saturate to the value predicted by a microcanonical ensemble [3, 4]. On the contrary, non-ergodic quantum systems violate ETH: there are many such examples, including localized systems [5–11], as well as models displaying Hilbert space fragmentation [12–19]. In the case of localization transitions, i.e. when a system undergoes a dynamical phase transition from an ergodic to a localized phase, the mechanism driving the phase change resides in the suppression of resonances across the transition [20].

In its traditional formulation, the notion of 'resonance' is related to other concepts such as 'level repulsion', 'anti-crossing', or 'avoided crossing' [21], and can be introduced as follows: the eigenvalues $E_{1,2}$ of a $2 \times 2$ real Hamiltonian $H$ can be approximated by its diagonal elements $\varepsilon_{1,2}$ provided its off-diagonal element $v$ is negligible compared to the difference between the diagonal elements, i.e., $v \ll \omega = \varepsilon_2 - \varepsilon_1$. Hence, if

$$v \gtrsim \omega, \tag{1}$$

the shifts $\Delta_{1,2} = E_{1,2} - \varepsilon_{1,2}$ are also greater than or of the order of $\omega$, and the sites are said to be 'in resonance' or 'hybridized' [22], meaning that the eigenstates occupy both sites about equally. Therefore, the presence of many resonances eases transport across different portions of the system, thus leading to ergodicity. Given this simple construction, it is tempting to generalize this idea to $N \times N$ matrices of arbitrary size $N$, saying that if there are $M$ sites $j = \{1, 2, ..., M\}$ such that $v_{0j} \gtrsim \varepsilon_j - \varepsilon_0$, then the zeroth site should be 'in resonance' with $\sim M$ other sites, and the corresponding eigenstate should have at least $\sim 1 + M$ relatively large components in the coordinate basis; this principle has found an extensive use not only in the studies of a single-particle localization [23–26] but also in the studies of the mesoscopic systems localization [27] and the many-body localization [28], including the ones considering the Anderson localization in the Hilbert space [29–32]. For example, provided the distribution of $v$ has a typical scale $v_{\text{typ}}$, one can correspondingly define a typical critical value of the energy difference $\varepsilon_2 - \varepsilon_1$ as $\omega_{\text{crit}} \sim v_{\text{typ}}$, meaning that all resonant sites should typically form a miniband of width $\omega_{\text{crit}}$ and, hence, the typical number of such resonant sites should be of the order of

$$M \sim \omega_{\text{crit}}/\delta_\varepsilon \sim v_{\text{typ}}/\delta_\varepsilon, \tag{2}$$

with $\delta_\varepsilon$ being the mean onsite energies spacing. This generalization provides the lower-bound estimation for the number of sites where the eigenstate has a relevant weight and it is usually

used in the thermodynamic limit $N \rightarrow \infty$ to distinguish between localized and delocalized states, giving rise to the necessary criterion $\lim_{N \rightarrow \infty} M(N) < \infty$ for the state to belong to the localized phase known as the Anderson localization criterion. Indeed, according to Anderson [5], the phase can be considered localized as long as the perturbation theory converges, and the condition $v_{\text{typ}} \ll \delta_\varepsilon$ is the convergence criterion based on the first-order perturbation theory.

However, the notion of resonance is sometimes referred also in relation to such concepts as 'fractal dimension,' 'support set volume,' and 'ergodic bubble', and here is why: in its extreme, a wave function $\psi$ having the majority of its weight on $\Omega$ sites can be imagined as having only $\Omega$ non-zero components of equal intensity $\psi(i)^2 = 1/\Omega$, leading to the participation entropy value $S = -\sum_i \psi(i)^2 \ln \psi(i)^2 = \ln \Omega$. In real-world situations, we can still introduce the support set volume $\Omega$ via its relation to entropy as, e.g., $\Omega = \exp(S)$, but then it would be as challenging to calculate it analytically as the entropy itself. On the other hand, from the analogy with the toy eigenstate having $\Omega$ equal non-zero components, it is clear that the ergodic volume $\Omega$ must be somehow related to the number of resonances $M$ introduced above. And, while $\Omega$ is indeed sometimes referenced as the 'number of resonances' [33], and thus it would be tempting to write $\Omega = 1 + M$, the actual relation is $\Omega \gtrsim M + 1$.

Hence, on the one hand, we have the easily calculable quantity $M$, which does not seem to have a clear relation to any observable in finite-size systems and, strictly speaking, can only be used in the thermodynamic limit to determine the localization transition. On the other hand, we have the ergodic volume $\Omega$, which is related to entropy and other commonly used observables but cannot be easily accessed analytically. On top of that, there is an intuition suggesting that there should probably be a more helpful relation between $\Omega$ and $M$ than the inequality above. In this work, we shed light on this relation.

The main result of the paper is a resonance criterion that doesn't make use of arbitrary thresholds, but rather is self-consistent and automatically avoids the issue. We also propose an ansatz for the wave function supported independently by phenomenological and microscopical considerations that, combined with the resonance criterion, allows us to compute observable quantities such as the participation entropy and the support set dimension. We then test the predictive power of our theory against numerical results on different types of Rosenzweig-Porter models.

The rest of the paper is organized as follows. In Sec. 2.1, we discuss a relation between resonance counting and the Jacobi diagonalization procedure, which leads us to the concept of dressed hopping. In Sec. 2.2, we argue that the dressed hopping is not the end of the story and propose a modification to the naive resonance condition given in the Introduction. In Sec. 3, we discuss the common problems of the resonance conditions introduced earlier and propose the phenomenological self-consistent criterion that solves them all; this is the main result of our paper. Then, in Sections 4 and 5, we test our self-consistent approach to resonance counting by numerically comparing it to the results of exact diagonalization for a range of random matrix models. Finally, we re-derive the previously studied resonance conditions from the exact microscopic size-increment equations in Sec. 6.1, provide an in-detail comparison of the resulting approximations with the phenomenological results and exact numerics in Sec. 6.2, and conclude the main part of the paper in Sec. 7.

## 2 Background and motivation

### 2.1 Resonance counting and Jacobi rotations

The reason why $M$ defined via Eq. (2) only provides the lower-bound estimation for $\Omega$ can be seen from the picture of Jacobi rotation [34]. For clarity, let us briefly introduce the Jacobi algorithm. The idea is very simple: given a symmetric matrix $H$, the iterative algorithm chooses an off-diagonal element $H_{ij}$ and applies a Givens rotation $U(i,j)$ on the $2 \times 2$ submatrix with elements $H_{ii}$, $H_{ij}$, $H_{ji}$, $H_{jj}$ in such a way that $[U(i,j)HU(i,j)^\dagger]_{ij} = [U(i,j)HU(i,j)^\dagger]_{ji} = 0$. This procedure, other than affecting the diagonal elements $H_{ii}$ and $H_{jj}$ causing level repulsion, will also affect all the matrix elements belonging to the same row or column of the decimated elements. This algorithm turns out to be effective in addressing properties in localized single-particle [35] and many-body quantum systems [20] and in well-thermalizing models [36]. Let us mention that different choices of the elements to decimate can be made. By choosing the current largest element one guarantees fast convergence, but in our setting it is useful to pick the off-diagonal element $H_{ij}$ for which $H_{ij}/(H_{ii} - H_{jj})$ is largest, representing the largest current resonance. This choice may not be the most efficient from the numerical point of view but, in some contexts, allows building successful analytic theories [23–26].

Consider a random matrix with a site having $M_0$ resonances in the coordinate basis; that would mean we need to perform at least $M_0$ rotations to eliminate these resonances and obtain the corresponding eigenstate. However, these $M_0$ rotations may create new resonances, and we will have to perform even more rotations involving our state to eliminate them. So, if one wanted to improve the lower bound for $\Omega$, they would need to consider the resonances not only on the first but also on the latter steps of the Jacobi diagonalization procedure. Assuming the subsequent rotations do not undo the preceding ones, the improved lower bound estimation then takes the form $\Omega \gtrsim 1 + \sum_i M_i$, where $M_i$'s are the numbers of resonances eliminated by the subsequent rotations.

The presented picture of Jacobi rotations can be employed in the following way. Consider a random matrix and pick a site; then, while performing Jacobi rotations, only apply them to the rest of the system, i.e., diagonalize the submatrix excluding the chosen site. After this sub-diagonalization, the hopping between our site and the rest of the system is expressed in the eigenbasis of the submatrix, i.e., it is now 'dressed' compared to the original 'bare' hopping. Thus, since the distribution of the dressed hopping elements contains information about the ergodic volume of the submatrix, the number of resonances counted using this distribution should provide a better estimation for $\Omega$ than the one utilizing the bare hopping distribution.

Another possible point of view on this construction is to consider the submatrix $H_N$ as the original system and the chosen site as the addition, increasing the size of the original system (see Fig. 1). So, if the addition of the new site does not disturb the eigenenergy $E_n$ of the original system too much, the corresponding eigenvector $|n\rangle$ does not redistribute too much of its weight to the newly added site. In contrast, if some other eigenenergy $E_k$ 'resonates' with the newly added site, this site would now likely be in the support set of the corresponding deformed eigenstate of the extended system. Thus, for a system with $M$ dressed resonances, we expect to see the newly added site in the support sets of $M$ submatrix-originating eigenstates. Hence, the eigenstate originating from the additional site must occupy at least $1 + M$ sites to be orthogonal to the rest of the eigenstates (it follows from counting the degrees of freedom). Therefore we get another lower-bound estimation for the support set volume, but this time, we expect it to be a much better bound than the one utilizing bare hopping instead of the dressed ones.

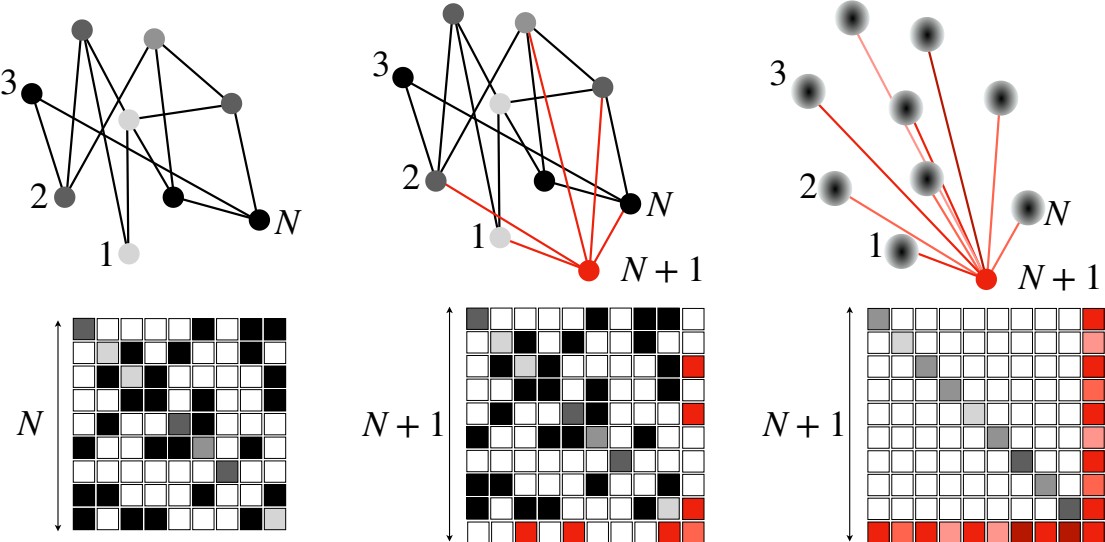

Figure 1: Pictorial representation of a site addition to a random graph (representing a random matrix). Different shading on nodes and edges represents different on-site energies and different hopping strengths. When a new site is added (right), a new row and column are added to the matrix, containing the hopping strength from the new site to the other sites (here represented in red shades).

## 2.2 Direct and indirect resonances

While the Jacobi rotations picture sheds some light on possible ways of improving the resonance counting procedure, there is still something missing from the picture. To see why, consider the Gaussian Rosenzweig-Porter (RP) ensemble [37] having independent uniformly distributed onsite energies $H_{ii} = \varepsilon_i \in [-w, w]$ and normally distributed hopping elements $H_{ij} = v_{ij}$ with zero mean and size-dependent variance $\langle v_{ij} \rangle^2 = N^{-\gamma}$: due to the Gaussian distribution of the bare hopping, the dressed hopping has precisely the same distribution as the bare one, as a linear combination of normally distributed random variables is also normally distributed. Still, since the traditional resonance condition (1) predicts the typical number of resonances $M$, Eq. (2), to be of the order of the ratio $v_{\text{typ}}/\delta$ with $\delta \sim \sqrt{\langle \text{Tr}\{H^2\}/N \rangle}/N \sim \max\{w/N, N^{-(1+\gamma)/2}\}$ now denoting the mean level spacing,[1] the scaling of the number of resonances $M \sim \min\{N^{1-\gamma/2}, N^{1/2}\}$ correctly predicts the Anderson localization transition $\gamma_{AT} = 2$ but severely underestimates the support set volume $\Omega \sim M$ in the delocalized phase, unable to correctly locate the ergodic-fractal transition $\gamma_{ET} = 1$ even in the thermodynamic limit.[2] The reason for that may be the resonance criterion itself as it is based on the analogy with the $2 \times 2$ matrices and the first-order perturbation theory. Hence, it should probably be modified when one talks about matrices of arbitrary size in the delocalized phase.

To find out the appropriate modification, notice a similarity between the site-addition picture from the end of Sec. 2.1 and the Thouless criterion of localization [41] based on comparing the eigenenergies' shift $\Delta$ induced by the boundary conditions change to the mean level spacing $\delta$. Indeed, the addition of the new site can be seen as an alteration of the boundary conditions for the original system, which may cause an indirect (and mediated by the new

---

[1]The change of the meaning of $\delta$ reflects the change of the physical picture in mind: while in Sec. 1 we were counting resonances between sites with certain onsite energies connected by bare hopping, here we count resonances between an arbitrary site and the eigenstates of the rest of the system, connected by the dressed hopping.

[2]Which is not a surprise given the aforementioned relation between this resonance condition (1) and the Anderson criterion of *localization* as opposed to the Mott's criterion of *ergodicity* [38–40].

site) resonance between close-in-energy unperturbed eigenstates, even in the absence of direct hopping between the two eigenstates [9]. Hence, the notion of the 'sufficient disturbance' to the original eigenenergies can be reconsidered: when a shift of the order of $\delta$ is enough for the state to hybridize with the newly added site, it looks like an overkill to require the shift $\Delta$ to be of the order of $\omega$ due to the direct resonance paradigm from Sec. 1.

To formulate the corresponding resonance condition mathematically, consider the extended $(N+1) \times (N+1)$ Hamiltonian $H_{N+1}$ and write the corresponding eigensystem equation in the block form as

$$
H_{N+1}|E\rangle = \left[ \begin{array}{c|c} H_N & |v\rangle \\ \hline \langle v| & \varepsilon \end{array} \right] \left[ \begin{array}{c} P_N|E\rangle \\ \hline \psi_E(\varepsilon) \end{array} \right] = E \left[ \begin{array}{c} P_N|E\rangle \\ \hline \psi_E(\varepsilon) \end{array} \right] = E|E\rangle .
\tag{3}
$$

In the above expression $H_N$ is the Hamiltonian of the original $N \times N$ system, $|v\rangle$ is a hopping column vector connecting the new site to the original system, $\varepsilon$ is the new site's onsite energy, $E$ and $|E\rangle$ are the extended Hamiltonian's eigenenergy and eigenstate, $P_N$ is a projector to the original system's Hilbert space, and $\psi_E(\varepsilon)$ is the eigenstate's amplitude on the new site, i.e., $|\psi_E(\varepsilon)|^2 = \langle E|(\mathbb{I} - P_N)E\rangle$, with $\mathbb{I}$ being the identity matrix. Then, acting in the spirit of Gaussian elimination, i.e., expressing $\psi_E(\varepsilon)$ from the eigensystem equation as

$$
\psi_E(\varepsilon) = \frac{\langle v|P_N|E\rangle}{E - \varepsilon}
\tag{4}
$$

and substituting it to the equation $H_N P_N |E\rangle + \psi_E(\varepsilon)|v\rangle = E P_N|E\rangle$, one finds that $P_N|E\rangle$ satisfies the (nonlinear) eigensystem equation $H_{eff}(E)P_N|E\rangle = E P_N|E\rangle$ with the effective Hamiltonian

$$
H_{eff}(E) = H_N + V_{eff}(E) = H_N + \frac{|v\rangle\langle v|}{E - \varepsilon} .
\tag{5}
$$

Hence, after linearizing the equation by substituting $E$ with the original Hamiltonian's eigenenergy $E_n$, $H_N|n\rangle = E_n|n\rangle$, we get from the perturbation theory for the linearized effective Hamiltonian $H_{eff}(E_n)$ that

$$
E - E_n = \Delta_n \sim \langle n|V_{eff}(E_n)|n\rangle \sim v_n^2/\omega_n ,
\tag{6}
$$

where $v_n = \langle n|v\rangle$ is the hopping vector's component in the eigenbasis $|n\rangle$ of $H_N$, and $\omega_n = E_n - \varepsilon$ is the difference between the original system's eigenenergy under consideration and the new site's onsite energy. Then, the new Thouless-inspired resonance condition takes the form

$$
v_n^2 \gtrsim \min\{\omega_n^2, \omega_n \delta\} ,
\tag{7}
$$

where the term $\omega_n^2$ in the r.h.s. appears due to the necessity of counting also the direct resonances between the newly added site and the subsystem's eigenstates having the closest eigenenergies to the site's onsite energy. In contrast to Eq. (1), the resonance condition (7) applied to the RP model with $1 < \gamma < 2$ predicts the number of resonances to scale as $M \sim \omega_{crit}/\delta \sim v_{typ}^2/\delta^2 \sim N^{2-\gamma}$ and gives the correct ergodic-fractal transition threshold at $\gamma_{ET} = 1$ ($M \propto N$, the volume law) as well as correct Anderson localization transition at $\gamma_{AT} = 2$ ($M \propto N^0$, finite support).

The reason why one should include the indirect resonances in the picture and relate the total number of all such resonances to the support set volume is the same as in Sec. 2.1, i.e., it is justified by counting the degrees of freedom. The only change is in the definition of the 'sufficient disturbance' of the original eigenenergies: if the site's addition can cause two eigenenergies of the original system to collide and the corresponding eigenstates to hybridize, it is reasonable to expect this site to be a part of their support sets.

## 3   Self-consistent resonance condition

While the resonance condition (7) looks more grounded than Eq. (1), they both have several problems in common. One of the problems is the threshold problem: what does '$\gtrsim$' actually mean? Without answering this question, one can only estimate the scaling of $M$ but cannot unambiguously compute the prefactor: while deriving Eq. 7, should we define the event of resonance as $\Delta > \delta$, or as $\Delta > \delta/2$, or as anything else? Moreover, the prefactor itself has little value unless the finite-size number of resonances $M$ is linked to some measurable observable, and this is the second common problem of resonance counting defined via conditions Eq. (1) and Eq. (7): given the value of $M$, how to calculate, e.g., the participation entropy $S$? To answer these questions, notice that, so far, the notion of resonances has always been related to the energy spectrum and eigenenergies shifts, while the target observables like $\Omega$ or $S$ are the properties of the wavefunctions. Hence, it seems reasonable to redefine the notion of resonances such that it would be directly linked to the wavefunctions' shape, which is what we do in the present section.

Consider the exact expression (4) for the occupation of the newly added site; after approximating $P_N |E\rangle$ by the unperturbed eigenstate $|n\rangle$ of $H_n$, we obtain the perturbation theory result for the occupation as

$$\psi_n(\varepsilon)^2 \sim \frac{v_n^2}{(E_n - \varepsilon)^2} = \frac{v_n^2}{\omega_n^2}. \tag{8}$$

For each particular realization of the random Hamiltonian under consideration, the approximation (8) may or may not hold independently of the phase our system is in as the approximation's applicability is only related to the very values of $v_n^2$ and $\omega_n^2$; for more detailed discussion of this fact, see Sec. 6.1. In other words, while $v_n^2/\omega_n^2$ is small enough, the perturbation theory works, but it breaks down if this ratio becomes larger than some threshold. The threshold is there to omit the divergence of $v_n^2/\omega_n^2$ at small $\omega$'s, i.e., due to the normalization condition, and the dominant contribution to the normalization of the wavefunction is commonly attributed to the support set [42] consisting of strongly hybridized sites with roughly equal occupations, i.e., the ergodic bubble or the head of the wavefunction; here and below, the terms "support set", "ergodic bubble", and the "wavefunction's head" will be used interchangeably due to their synonymous meaning. Based on this idealized picture, we conclude that the occupation of the newly added site by the eigenstate $\psi_n$ should look like

$$\psi_n(\varepsilon)^2 \sim \begin{cases} \psi_{head}^2, & v_n^2/\omega_n^2 > C/\Omega \\ v_n^2/\omega_n^2, & v_n^2/\omega_n^2 < C/\Omega \end{cases}, \tag{9}$$

where $\psi_{head}^2$ is distributed as components of a fully ergodic eigenstate, $\Omega$ is the number of sites in the support set,[3] and $C$ is the total weight of the state concentrated in its head, i.e., $C = \Omega \langle \psi_{head}^2 \rangle$ (see Fig. 2 for a visual representation of $\psi^2$). From this point of view, the probability of a resonance can be unambiguously defined as the probability for the newly added site to become part of the perturbed eigenstate's head, and the corresponding resonance condition takes the form

$$v_n^2 > \omega_n^2 C/\Omega, \tag{10}$$

where $C$ and $\Omega$ should be self-consistently determined from the equations

$$C = 1 - (N + 1 - \Omega)\langle v_n^2/\omega_n^2 \rangle_{tail}, \quad \Omega = 1 + NP(\Omega, C); \tag{11}$$

here, $P(\Omega, C)$ is the probability for Eq. (10) to hold (i.e., the probability of resonance), while the averaging $\langle ... \rangle_{tail}$ in the expression for $C$ is calculated only over the values of the ratio

---

[3]Not to be confused with the relation $\Omega = \exp(S)$ briefly mentioned in Sec. 1; here and below, $\Omega$ has a similar physical meaning but a more complicated relation to $S$ which is discussed in the present section.

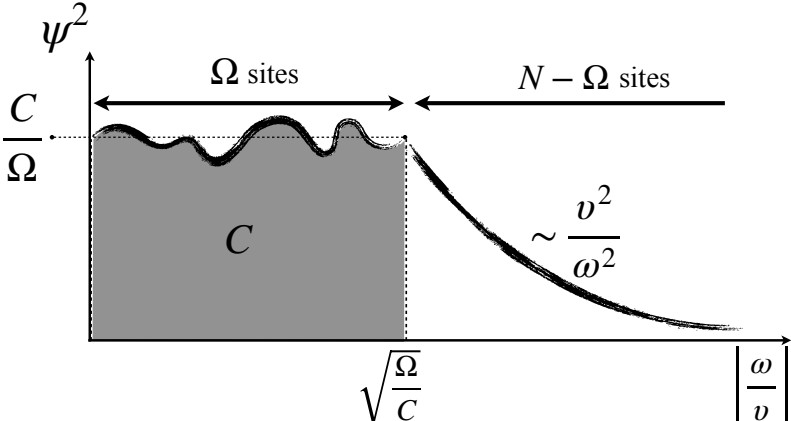

Figure 2: Visual representation of the wavefunction according to Eq. Eq. (9). It can be represented as split into two parts: the head, having support on $\Omega$ sites and taken as a Haar random vector, and the tail, where there are no resonances and one can use the perturbation theory expression.

$v_n^2/\omega_n^2$ which do not exceed $C/\Omega$. The above equations are easily obtained: the number $\Omega$ of sites in the head is simply given by the number of resonances, $N$ times the probability of resonance, plus '1' standing for the newly added site.[4] On the other hand, one minus the average value of $\psi_n^2$ in the tails times the number of sites in the tails gives the total weight $C$ in the head. A microscopic derivation of Eq. (10) can be performed via the secular equation and we discuss it in Sec. 6, where we also highlight the connection with the resonance condition Eq. 7.

As one can see, the self-consistent resonance condition does not have the threshold problem as the threshold is determined self-consistently and has a clear physical meaning. Indeed, equating the l.h.s. and the r.h.s. of Eq. (10) and using Eq. (6), we see that, in the borderline case between resonant and non-resonant, the energy shift $\Delta$ is of the order of $\omega_{\text{crit}}C/\Omega \sim C\delta$; so, the value of the threshold is equal to $C$, the total eigenstate's weight attributable to the resonant sites. However, since the definition of $\omega_{\text{crit}}$ requires the existence of a typical scale of the dressed hopping distribution, one should not understand this threshold picture too literally but rather just use the self-consistent approach to resonance counting as described above.

Another advantage of this approach is its immediate connection to measurable observables like participation entropy. Indeed, provided all sites of the system are statistically equivalent, one can readily calculate such quantities using Eq. (9) as the ansatz for the wavefunction components. In this case, the distribution of $\psi_{head}^2$ can be modeled by, e.g., the beta distribution, as it is the distribution of the components of the Gaussian Orthogonal/Unitary/Symplectic Ensemble Hamiltonian's eigenstates, see App. A.

To conclude the Section and for further convenience, we provide here the analytical expressions for the probability of resonance $P(\Omega, C)$, the mean tail's occupation $\left\langle v_n^2/\omega_n^2 \right\rangle_{tail}$ entering the equations Eq. (11), and the participation entropy calculated using the ansatz Eq. (9). For simplicity, consider the eigenstates in the middle of the spectrum and put $E_n = 0$ so that $|\omega_n| \sim |\varepsilon|$; then, assuming the onsite energies to be uniformly distributed between $\pm w$, we get

---

[4]More concretely, this '1' comes from the fact that the approximation (8), being indexed by $n = 1...N$, can approximate at most $N$ out of the $N+1$ eigenstates of the extended system as it cannot approximate the eigenstate adiabatically connected with the one localized on the new site in the limit $v_k \to 0$ for all $k$. By the adiabatic continuity, this special eigenstate always has (one of) the largest occupation(s) of this site and hence always counted as a part of the head. For more details on this, see Sec. 6.1 and Fig. 12b.

the probability for $v_n^2$ to be larger than $\omega_n^2 C/\Omega$ as

$$P(\Omega, C) = \int_0^w \frac{\mathrm{d}\omega}{w} \int_{\omega^2 C/\Omega}^\infty p_{v^2}(\xi)\mathrm{d}\xi = 1 - \int_0^{w^2 C/\Omega} \mathrm{d}\xi p_{v^2}(\xi)\left(1 - \sqrt{\frac{\xi\Omega}{w^2 C}}\right), \qquad (12)$$

where $p_{v^2}(\xi)$ is the probability distribution function (PDF) corresponding to the distribution of the dressed hopping elements squared. The mean tails occupation can be obtained similarly and takes the form

$$\left\langle \frac{v_n^2}{\omega_n^2} \right\rangle_{tail} = \int_0^{w^2 C/\Omega} \mathrm{d}\xi p_{v^2}(\xi) \int_{\sqrt{\xi\Omega/C}}^w \frac{\mathrm{d}\omega}{w} \frac{\xi}{\omega^2} = \int_0^{w^2 C/\Omega} \mathrm{d}\xi p_{v^2}(\xi)\left(\sqrt{\frac{\xi C}{w^2\Omega}} - \frac{\xi}{w^2}\right). \quad (13)$$

Finally, the specific mean tail's participation entropy $s_{tail} = \left\langle -(v_n^2/\omega_n^2)\ln(v_n^2/\omega_n^2)\right\rangle_{tail}$ becomes

$$s_{tail} = -\int_0^{w^2 C/\Omega} \mathrm{d}\xi p_{v^2}(\xi)\left(\sqrt{\frac{\xi C}{w^2\Omega}}\left(\ln\left(\frac{C}{\Omega}\right) - 2\right) + \frac{2\xi}{w^2}(1 + \ln w) - \frac{\xi \ln \xi}{w^2}\right), \qquad (14)$$

and the total participation entropy for the beta-distributed head components takes the form

$$S = \Omega s_{head} + (N + 1 - \Omega)s_{tail} = C\left(H(\Omega/2) - H(1/2)\right) - C\ln(C) + (N + 1 - \Omega)s_{tail}, \quad (15)$$

where $H(\Omega/2)$ is the Harmonic number, and $s_{head}$ is calculated in App. A.

## 4   Analytical study of the Gaussian Rosenzweig-Porter model

Now, after having the improved resonance condition at our disposal, let us try it on the Gaussian Rosenzweig-Porter model, which is defined as

$$H_{\mathrm{GRP}} = H_0 + V, \qquad (H_0)_{ij} = \epsilon_i \delta_{ij}, \ \epsilon_i \in [-w, w], \qquad V = N^{-\gamma/2}H_{GOE}, \qquad (16)$$

where the elements of $H_{GOE}$ are i.i.d. Gaussian r.v.s, with zero mean and unit variance. This model has a non-trivial phase diagram, displaying, irrespectively of the value of $w$, an ergodic phase for $\gamma < 1$, a fractal phase for $1 < \gamma < 2$, and a localized phase for $\gamma > 2$ [37, 43]. The main advantage of this model for our purposes is that the dressed hopping distribution is known exactly and, as it has already been mentioned in Sec. 2.2, coincides with the distribution of the bare hopping. Therefore we can directly substitute the PDF of the normal distribution to Eqs. (12) and (13), numerically solve Eqs. (11) and, using Eq. (9) with the beta-distributed head (see App. A), compute the participation entropy $S(N)$ and the related quantities such as the support set dimension

$$D(N) = \frac{S(N)}{\ln(N)} \qquad (17)$$

and the corresponding beta-function (see Sec. 4.1 for details)

$$\beta(N) = \frac{\mathrm{d}\ln(D)}{\mathrm{d}\ln(N)}; \qquad (18)$$

the results are depicted in Fig. 3, and we discuss in some more detail the properties of the $\beta$-function in Sec. 4.1, as it is a new prediction showing some unexpected features. The code that performs the analysis just described and that we used for producing the results reported in the next paragraphs can be found in the GitHub repository in Ref. [44].

Let us mention that the above definition of the support set dimension in Eq. (17) (which is just the fractal dimension $D_q$ with $q = 1$) is commonly used in the literature [43, 45], but different definitions are possible, e.g. the differential support set dimension $\mathcal{D}(N) = \mathrm{d}S(N)/\mathrm{d}\ln(N)$, which was used in Refs. [33, 46] for addressing the renormalization group flow in the Anderson model. The advantage of $\mathcal{D}(N)$ consists of having milder finite-size effects, albeit being numerically less stable, because of the presence of the derivative. In the thermodynamic limit, the two quantities are equivalent, and here, since we do not aim at reducing the finite-size effects but at understanding them, we choose to work with Eq. (17) for better numerical stability and easier comparison with the literature on Rosenzweig-Porter models. As one can see, the self-consistent resonance condition (10) not only correctly reproduces the thermodynamic limit phase diagram but also qualitatively captures the finite-size effects.

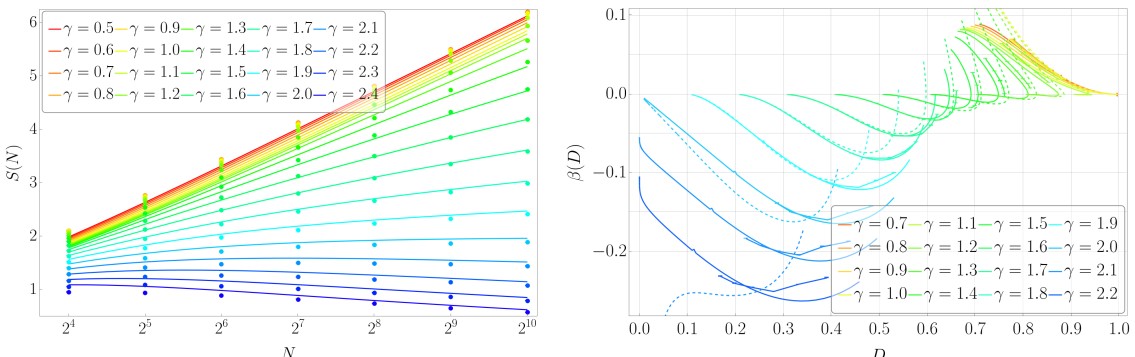

Figure 3: The participation entropy (left panel) and the support set dimension's beta function (right panel) for the Gaussian RP model with $w = 1$ obtained using exact diagonalization (points and broken lines) and the self-consistent resonance criterion (continuous curves). For the latter, the size varies from $N = 2^4$ to $N = 2^{100} - 2^{300}$, depending on $\gamma$. Also, the dashed lines on the right panel show another analytical prediction for the same quantity derived in App. B based on the ideas from Ref. [47]; for further discussion of this result, see App. B.

An intriguing and somehow unexpected behavior of the total head's weight $C$ is given in Fig. 4: as one can see, this quantity exhibits substantial finite size effects which can be observed even at $N = 2^{100}$. In addition to that, $C$ has two limiting thermodynamic values corresponding to the ergodic/localized phases ($C = 1$) and non-ergodic delocalized ($C = 0.5$) phases, while, at the transition points $\gamma_{AT} = 2$ and $\gamma_{ET} = 1$, $C$ saturates at intermediate $w$-dependent values.

## 4.1 The beta-function of the Gaussian Rosenzweig-Porter model

Let us discuss a bit more in detail the analytical prediction for the $\beta$-function of the Gaussian RP model. First of all, as we already emphasized, it matches the numerical results at a finite size, and therefore, its predictions are reliable. It is natural to compare it with the results obtained for the Anderson model on Random Regular Graphs [33], in finite dimension [46], and in many-body localization [48], as there are interesting differences.

Let us first mention some basic properties. The support set dimension $D$ is bounded, $0 \leq D \leq 1$, while it is not true in general at finite size for the differential support set dimension $\mathcal{D}$; see, e.g., the behavior of the participation entropy for $\gamma > 2$ in the left panel of Fig. 3, where its slope is negative and, hence, $\mathcal{D} < 0$. Also, at finite size, the flow curves have to have an infinite slope when approaching the $\beta = 0$, in order to cross it for a finite value of the system size. If, instead, the slope is finite, the $\beta = 0$ line can be reached only in the thermodynamic limit. We can see in Fig. 3 this feature. Let us also remark that, as expected,

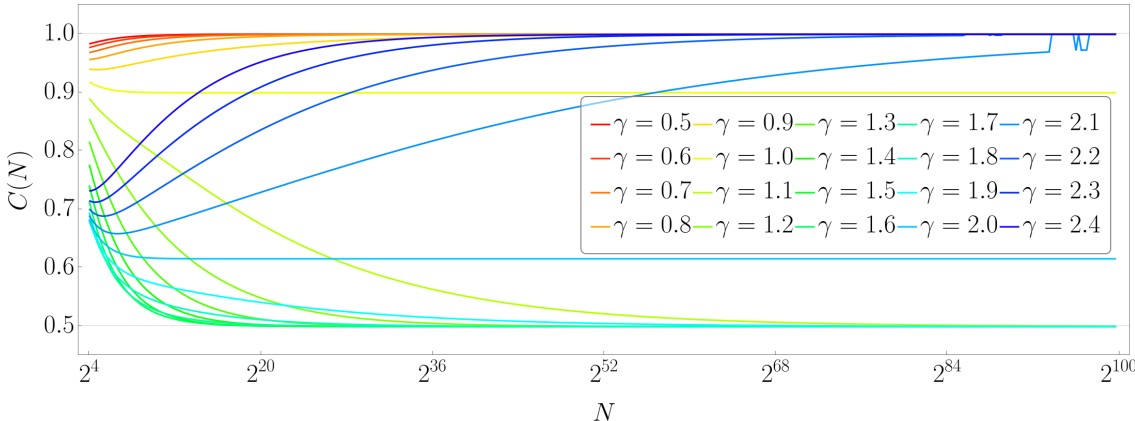

Figure 4: The total head's weight $C$ according to the solution of Eq. (11) for $w = 1$. Given that the value of $C$ can be associated with the threshold value for the indirect resonance condition (7), we can conclude that Eq. (7) should also work fairly well even without the self-consistent equations as $0.5 \lesssim C \lesssim 1$.

for $\gamma \in (1, 2)$ the termination point of the RG flow occurs at $0 < D < 1$, signaling the presence of the fractal phase. For $\gamma > 2$, the fractal dimension in the thermodynamic limit vanishes; interestingly enough, the $\beta$-function value in the thermodynamic limit is negative, and it's magnitude increases with $\gamma$. More specifically, estimating this limiting value directly from the perturbation theory (which works very well in the localized phase with $\gamma > 2$), we get

$$S(N) = N \left\langle -\psi_n(\varepsilon)^2 \ln\left(\psi_n(\varepsilon)^2\right) \right\rangle \sim N \left( c \frac{v_{typ}}{w} - 2 \int_{v_{typ}}^{\infty} d\omega \frac{v_{typ}^2}{\omega^2} \ln \frac{v_{typ}^2}{\omega^2} \right) \propto N^{1-\gamma/2}, \quad (19)$$

where $c \propto N^0$ stands for the head contribution corresponding to the (unlikely) event of $|E_n - \varepsilon| = \omega \lesssim v_{typ} = N^{-\gamma/2}$ happening with the probability $v_{typ}/w$. This suggests the presence of a line of fixed points ($D = 0, \beta = 1 - \gamma/2$) in the localized phase, see Fig. 3, similarly to what happens in the Anderson model on RRGs [33]. Let us also notice that the analytical predictions of Ref. [47] are not valid in the localized regime (see also App. B), as visible from the dashed lines.

Let us now discuss some differences with the RG flow in the Anderson model. Our aim is not to draw connections between these models, as there is no theoretical reason for them to have similarities in a renormalization group sense, but just to describe the differences the models display. In Refs. [33, 46] the authors describe the full renormalization group flow for the differential support set dimension in the Anderson model, both on Random Regular Graphs and in finite dimensions. From the behavior of the $\beta$-function, the authors verified the one-parameter scaling hypothesis in both cases in the ergodic regime. In the present case, there cannot be one-parameter scaling in the fractal phase, by definition. But there is another interesting distinction: in the Anderson model, in the ergodic phase, the differential support set dimension displays a minimum and then saturates to the ergodic value $\mathcal{D} = 1$ when the system size is increased. In the Gaussian RP model, the opposite happens in the fractal phase with the plain support set dimension $D$, which has a maximum at very small sizes before flowing to the saturation value.

# 5 Resonance counting in other Rosenzweig-Porter models

In this Section, we extend the results previously shown for the Gaussian Rosenzweig-Porter model to other random-matrix ensembles, still displaying a localization transition. However, for non-Gaussian cases, we are not able to compute explicitly the distribution of dressed hoppings, and thus we need to solve numerically the self-consistent resonance condition (10). Since our ultimate goal is to address the properties of the Anderson model, after a consistency check we will focus on Rosenzweig-Porter models resembling features of the Anderson model on random graphs.

Let us briefly explain how we numerically solve the self-consistent equations from Sec. 3. The main goal of the numerical solution is that of having the correct distribution of dressed hopping, since the analytical computation of the distribution lies beyond the scope of this paper. To do so, we numerically compute the exact eigenvectors $|n\rangle$ and eigenvalues $E_n$ of many samples of random matrices. We then collect the corresponding samples of dressed hoppings $v_n = \langle n|v\rangle$ by independently sampling the new hopping vectors $|v\rangle$, and the energy differences $\omega_n = E_n - \epsilon$ by independently sampling new onsite energies $\epsilon$'s, ultimately obtaining a collection of $\psi_n^2 = v_n^2/\omega_n^2$. Once a sufficiently large sample (let us say of size $m$) of $\psi_n^2$'s is collected, we sort it in ascending order and, while iterating through the sample with the index $k$, we compute the "current" $P$, $\Omega$, and $C$ as

$$P(k) = 1 - k/m, \quad \Omega(k) = 1 + NP(k), \quad C(k) = 1 - (N + 1 - \Omega(k))\langle\psi_j^2\rangle_{j<k}, \quad (20)$$

where $k$ is the current position in the sample and $\langle...\rangle_{j<k}$ represents the average of the elements up to the $k$-th. For each $k$, we check whether $\psi_k^2 > C(k)/\Omega(k)$; when this condition is satisfied for the first time, we compute the entropy in the tails as

$$s_{tails} = -\frac{1}{k}\sum_{j=1}^{k}\psi_k^2 \log\psi_k^2 \quad (21)$$

and use the current values of $\Omega$ and $C$ to obtain the expression for the total entropy according to Eq. (15). The code that performs the analysis just described and that we used for producing the results reported in the next paragraphs can be found in the GitHub repository in Ref. [44].

## 5.1 A further check on Gaussian Rosenzweig-Porter

As a first check, we compute numerically the probability of resonances, and consequently the participation entropy, for the Gaussian Rosenzweig-Porter model (that we solved analytically in Sec. 4). As it should, the match for participation entropy and support set dimension between exact diagonalization and numerical resonance counting due to the self-consistent resonance condition is remarkably good, as shown in Fig. 5.

## 5.2 The log-normal Rosenzweig-Porter model

Let us now take a more complicated Rosenzweig-Porter model, called log-normal Rosenzweig-Porter model. It is defined by the matrix ensemble

$$H_{\text{LNRP}} = H_0 + V, \qquad (H_0)_{ij} = \epsilon_i\delta_{ij}, \qquad p_V(v) \propto \frac{1}{|v|}\exp\left\{-\frac{\ln^2(|v|/N^{-\gamma/2})}{2p\ln(N^{\gamma/2})}\right\}, \quad (22)$$

where $\epsilon_i$ is, again, uniformly distributed, $\epsilon_i \in [-w, w]$. Setting $p = 1$, one obtains a phase diagram according to which the system is ergodic for $\gamma < 4$ and localized for $\gamma > 4$, with $\{p = 1, \gamma = 4\}$ being a tricritical point on the phase diagram in the variables $\{p, \gamma\}$ [43, 49].

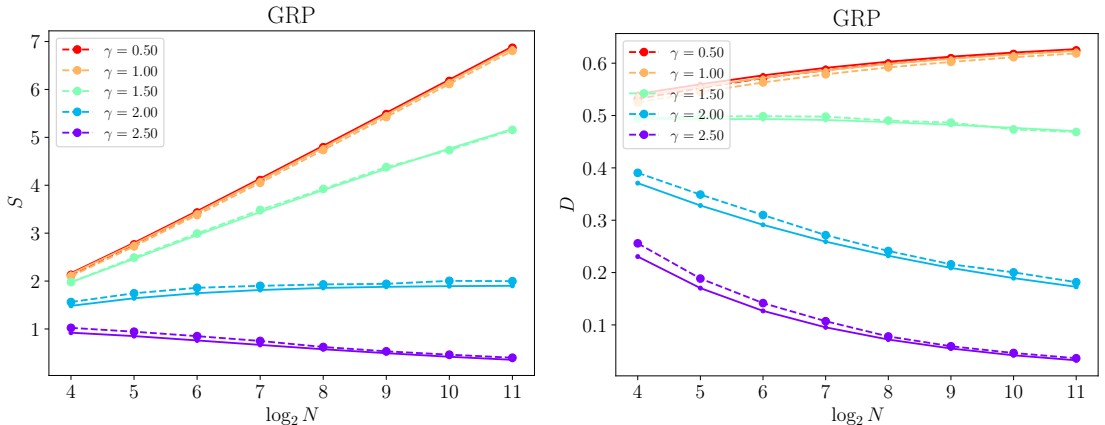

Figure 5: Comparison between exact diagonalization (solid lines) and analytical prediction (dashed) for the Gaussian Rosenzweig-Porter model. (Left) Participation entropy. (Right) support set dimension.

The interest in the log-normal RP model resides in its similarity with the Anderson model on Random Regular Graphs. Indeed, it has been shown that the distribution of the effective long-range hopping in the Anderson model on RRGs is approximately given by the log-normal distribution with $p = 1$ [38].

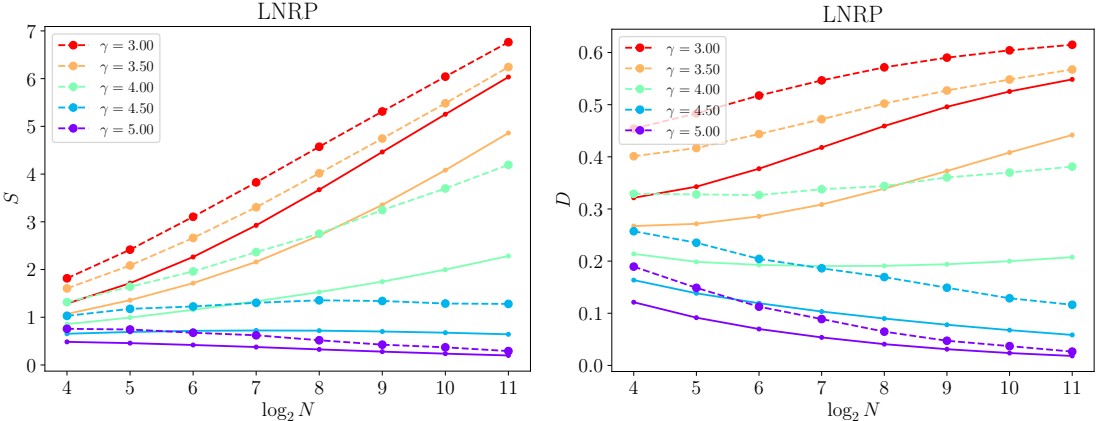

Figure 6: Comparison between exact diagonalization (solid lines) and analytical prediction (dashed) for the log-normal Rosenzweig-Porter model. (Left) Participation entropy. (Right) support set dimension (17).

We show in Fig. 6 the comparison between the analytical prediction following from Eq. (10) and the numerical results coming from exact diagonalization. We can notice that the qualitative behavior of the numerical and analytical results is the same, and, despite the numerical values being different (which is due to the fact that the correct distribution ansatz for $\psi_{head}^2$ is unlikely to be as simple as in the Gaussian RP case), the numerical and analytical curves tend to approach each other as the size grows, hinting that the physical behavior is correctly captured by our analytical description also in this case.

In particular, notice that, for $\gamma = \gamma_c = 4$, the analytical results for $D$ display a minimum as the numerics do, for roughly the same values of the systems size. The fact of the minimum's existence is non-trivial. Indeed, as the function $D(N; p, \gamma)$ is expected to be an analytic and, hence, smooth function of all the parameters at finite system sizes, the minimum cannot immediately disappear at $\gamma > \gamma_c$ or $\gamma < \gamma_c$, implying the existence of a vast range of possible

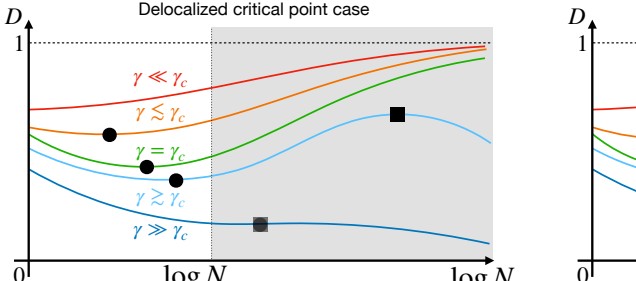 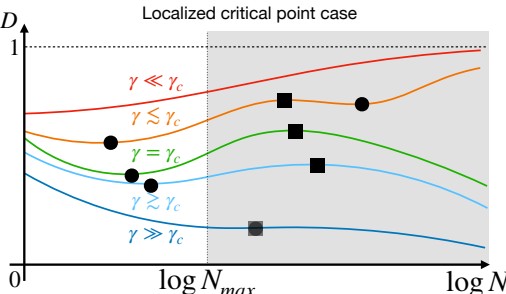

Figure 7: Sketch of the dependence of the support set dimension $D$ on the system size $L = \log N$. In both cases, $\log N_{max}$ represents the maximum system size available from exact-diagonalization (see Fig. 6), while the part in the shaded region is speculative and represents the object of our discussion. (Left) As suggested by the numerical data and from previous results, for $\gamma = \gamma_c$ the model is ergodic in the thermodynamic limit (differently from the RRG). This leads to the presence of a maximum in $D$ at large sizes, as explained in the main text. For some $\gamma$, minimum and maximum merge in a saddle. (Right) Assuming the critical point is localized, as a consequence of continuity in $\gamma$ there must be an additional maximum and minimum for $\gamma \lesssim \gamma_c$, which is harder to comprehend.

behaviors of the support set dimension at larger sizes. Assume, for example, that the critical point of the LN-RP model is localized; this assumption seems reasonable as the model is claimed to be a proxy for the Anderson model on RRGs, which is localized at its critical point. However, this would imply that the critical curve $D(N)$ must have at least one more extremum at larger sizes – a maximum. Moreover, by the function's analyticity, this maximum would also be present in the close-to-criticality ergodic phase, i.e., at $\gamma < 4$, resulting in the support set dimension $D(N)$ having at least three extrema at large but finite sizes in this phase, see the right panel of Fig. 7. On the other hand, if one would assume the critical point to be ergodic or at least fractal with the limiting support set dimension $D(N = \infty)$ depending on $w$, it would be possible to avoid the introduction of an additional extrema in the ergodic phase; the localized one though would still have to have at least two extrema, with the maximum emerging from $N = \infty$ as $\gamma > \gamma_c$ deviates from its critical value, see the left panel of Fig. 7. In fact, there are indications that the critical point of the log-normal Rosenzweig-Porter model is indeed delocalized; it can be inferred from, e.g., the self-consistent graphical solution for the LN-RP limiting support set dimension presented in [43], though the tricritical point lies at the very boundary of the graphical methods' applicability.[5] This fact, together with the complexity of the finite-size effects the model must show to have the tricritical point localized, poses questions about the extent of similarities between the LN-RP model and the Anderson model on RRG.

## 5.3 The Bernoulli Rosenzweig-Porter model

As a further step in the direction of the Anderson model on RRG, let us introduce the Bernoulli Rosenzweig-Porter model. It essentially consists of an Anderson model on an Erdos-Renyi graph [43], in the sense that, given $N$ sites, each of them is connected to another one by a unit hopping with an assigned probability that, in our case, is $K/N$; this also motivates the name "Bernoulli Rosenzweig-Porter model". This choice allows us to have, on average, connectivity

---

[5]In [43], the equation (51) defines a quantity $c$, related to the support set dimension as $D = 1 - c$, which vanishes as one approaches the tricritical point from either direction on the phase diagram, Figure 11, meaning that the limiting value of $D$ at this point is 1, corresponding to the ergodic phase.

$K$, as in the RRG, with the advantage of having the possibility of adding a single site without having to reshuffle the full adjacency matrix of the graph. On the other hand, the graph is not strictly regular, but only on average.

The Hamiltonian is therefore

$$H_{\text{ER}} = H_0 + V, \qquad (H_0)_{ij} = \epsilon_i \delta_{ij}, \qquad \epsilon_i \in [-w, w], \tag{23}$$

with $V$ being the adjacency matrix of an Erdos-Renyi graph, i.e., with $V_{ij} = 1$ if $i$ is connected to $j$ and zero otherwise. To the best of our knowledge, the Bernoulli Rosenzweig-Porter model has never been introduced before (although similar models have been considered, e.g. Ref. [50]), so we do not know its properties such as the position of the localization transition precisely. We expect it to be comparable with the value $W_c = 18.17$ of the RRG [51]. Our goal here is to test our analytical approximations against the exact numerical results, and the comparison is shown in Fig. 8.

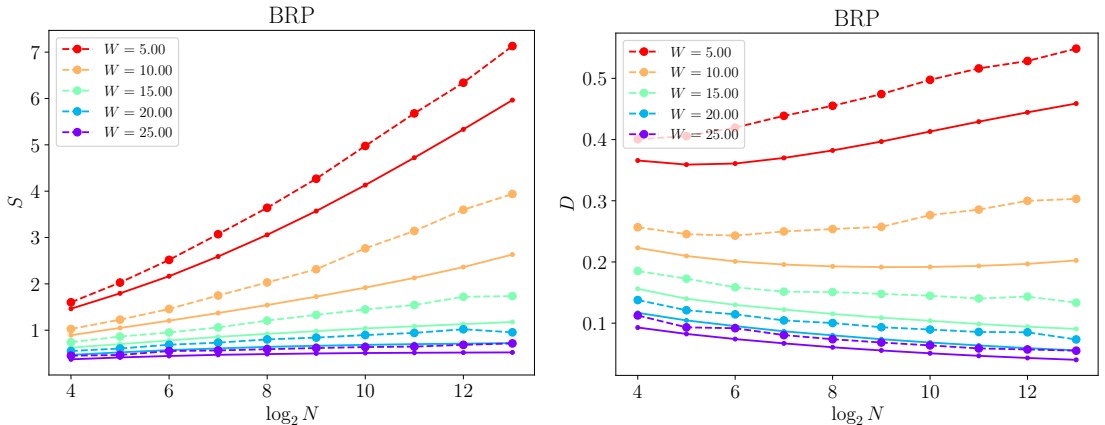

Figure 8: Comparison between exact diagonalization (solid lines) and analytical prediction (dashed) for the Bernoulli Rosenzweig-Porter model. (Left) Participation entropy. (Right) support set dimension.

Once again we can see that the qualitative picture is correctly captured by our resonance criterion, despite the quantitative difference between numerical and analytical results. This disagreement is most probably a consequence of the ansatz for the heads of the eigenstates, which might be not optimal for sparse matrices. However, the main goal is achieved also in this case: our self-consistent resonance criterion captures the finite-size effects qualitatively correctly, and we hope to find the quantitative correspondence also improving with size when the larger sizes become accessible to exact diagonalization.

## 6 Microscopic approach to resonance criteria

The resonance conditions Eq. (1) and Eq. (7) were introduced from the energy spectrum point of view, while the self-consistent condition (10) was based directly on the spatial eigenstates' configuration. Still, both Eq. (7) and Eq. (10) can correctly predict the full phase diagram in the thermodynamic limit and, for the Gaussian RP model, even give qualitatively similar values of $\Omega$.[6] In this section, we take a step back and analyze the resonance conditions described

---

[6]It can be inferred from Fig. 4 and the applicability of the threshold picture to the Gaussian RP model, suggesting that the threshold $C$ is of the order of 1 and does not significantly change with size, implying a rough equivalence between (7) and (10) in this particular case.

previously from a microscopic viewpoint, performing a careful asymptotic analysis of the exact equations for eigenenergies and eigenstates of the extended system. This allows us to give an even more solid motivation to our results.

## 6.1 Asymptotic analysis of the exact size-increment equations

To start with, suppose we know everything about the Hamiltonian $H^0$ and the arbitrary (not necessarily small) perturbation $V$; our task is to find the eigensystem of $H = H^0 + V$. It can be done as follows: first, we rewrite the eigensystem equation $(H^0 + V)|E\rangle = E|E\rangle$ in the form

$$|E\rangle = G^0(E)V|E\rangle \tag{24}$$

with the resolvent $G^0(E)$ defined as $G^0(E) = (E - H^0)^{-1}$; second, we obtain the secular equation as $||G^0(E)V - \mathbb{I}|| = 0$. If $V$ is a rank-one matrix, e.g., $V = |g\rangle\langle g|$, it gives the well-known secular equation of the Richardson model [52–56], $\langle g|G^0(E)|g\rangle = 1$. If $V$ has rank two, $V = |u\rangle\langle v| + |v\rangle\langle u|$, the secular equation takes the form

$$\left\| \begin{matrix} G^0_{uv}(E) - 1 & G^0_{uu}(E) \\ G^0_{vv}(E) & G^0_{vu}(E) - 1 \end{matrix} \right\| = 0, \qquad \text{with } G^0_{uv}(E) = \langle u|G^0(E)|v\rangle. \tag{25}$$

If the $(N+1) \times (N+1)$ Hamiltonian $H^0$ represents the system of $N$ connected sites with eigenenergies $E_n$ together with one disconnected site with onsite energy $\varepsilon$, the Hamiltonian $H$ with

$$V = |v\rangle\langle\varepsilon| + |\varepsilon\rangle\langle v| \tag{26}$$

represents the system where this lonely site $|\varepsilon\rangle$ is connected to the rest of the system via the hopping vector $|v\rangle$. In other words, if we consider the connected $N \times N$ block of $H^0$ as the Hamiltonian $H_N$ of the original $N$-sites system, the Hamiltonian $H$ can be seen as the Hamiltonian $H_{N+1}$ of the extended system, and the exact secular equation Eq. (25) provides the way to study the evolution of the eigenenergies as we grow the system size site by site. The secular equation then takes a simpler form $G^0_{vv}(E) = 1/G^0_{\varepsilon\varepsilon}(E) = E - \varepsilon$, or, explicitly,

$$\sum_{n=1}^{N} \frac{v_n^2}{E - E_n} = E - \varepsilon, \tag{27}$$

where $v_n = \langle n|v\rangle$ is the component of the hopping vector $|v\rangle$ in the eigenbasis of $H_N$, i.e., the dressed hopping.

As one can see from Fig. 9, the equation (27) has $N + 1$ solutions for $E$, each of those lying in-between two neighboring eigenvalues $E_n$ of the original system. This observation suggests rewriting of Eq. (27) in the form

$$\Delta_k = v_k^2 \Big/ \left( E_k - \varepsilon + \Delta_k - \sum_{n \neq k} \frac{v_n^2}{\omega_{kn} + \Delta_k} \right), \tag{28}$$

where $\omega_{kn} = E_k - E_n$, and $\Delta_k = E - E_k$ is the new unknown. For any fixed $k$, there are $N + 1$ solutions for $\Delta_k$, as it should be; however, the goal behind this rewriting is not to find all roots for fixed $k$, but to find the least-absolute-value solutions for each $k$. Such solutions never exceed the corresponding level spacing and can be either of the order of the typical level spacing $\delta$ or smaller, fitting in Thouless's picture of eigenvalues shifts from Sec. 2.2. Thus, assuming $\Delta_k \ll \min\{\omega_{k,k+1}, \omega_{k,k-1}\}$, we can write an asymptotic version of Eq. (28) as

$$\Delta_k \sim \frac{v_k^2}{E_k - \varepsilon - \sum_{n \neq k} v_n^2/\omega_{kn} + \Delta_k(1 + \sum_{n \neq k} v_n^2/\omega_{kn}^2)} = \frac{v_k^2}{\omega_k + \Delta_k \Gamma_k^2}, \tag{29}$$

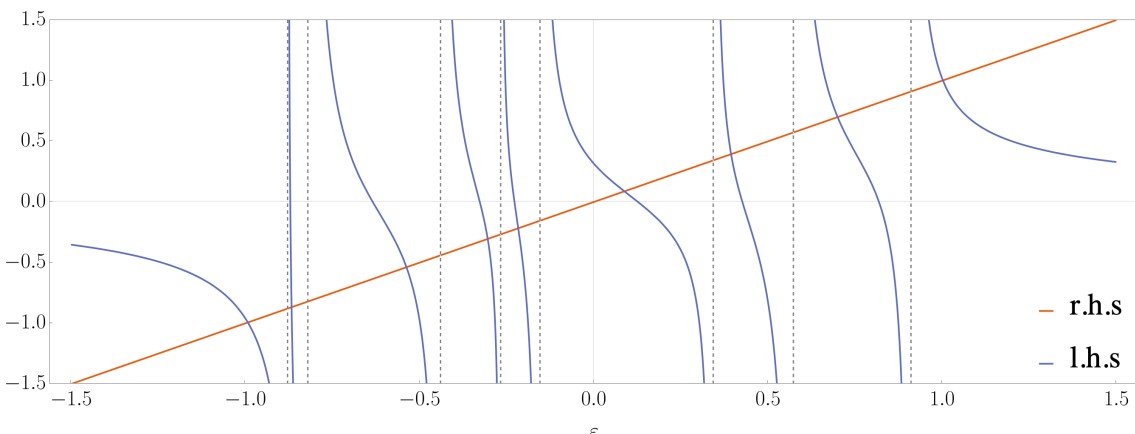

Figure 9: An example of a graphical solution to Eq. (27) with $\varepsilon = 0$ for the Gaussian RP model with $\gamma = 1.5$, $N = 8$, and $w = 1$; the red line is the equation's right-hand side, the blue curves represent the left-hand side, and their intersections represent the solutions, $E$. The vertical dashed lines mark the eight eigenenergies $E_n$ of the original system.

which leads to an easily solvable quadratic equation for $\Delta_k$, resulting in

$$\Delta_k \sim \text{sign}(\omega_k)\frac{\sqrt{\omega_k^2 + 4v_k^2\Gamma_k^2} - |\omega_k|}{2\Gamma_k^2}; \tag{30}$$

here, we chose the smallest $\Delta_k$ and defined

$$\omega_k = E_k - \varepsilon - \sum_{n \neq k} v_n^2/\omega_{kn}, \qquad \Gamma_k^2 = 1 + \sum_{n \neq k} v_n^2/\omega_{kn}^2. \tag{31}$$

Finally, we can substitute this approximate solution to the roughened version $\Delta_k \ll \delta$ of the above approximation's applicability condition $\Delta_k \ll \min\{\omega_{k,k+1}, \omega_{k,k-1}\}$ and obtain its explicit form as

$$v_k^2 \ll \Gamma_k^2\delta^2 + \omega_k\delta. \tag{32}$$

If, in addition, $\omega_k \gg \Delta_k\Gamma_k^2$, instead of Eq. (30) we can get

$$\Delta_k \sim v_k^2/\omega_k, \tag{33}$$

which is asymptotically correct provided both $v_k^2/\omega_k \ll \delta$ and $v_k^2/\omega_k \ll \omega_k/\Gamma_k^2$ hold, i.e.,

$$v_k^2 \ll \min\{\omega_k^2/\Gamma_k^2, \omega_k\delta\}. \tag{34}$$

Here, one may notice a similarity between the approximation applicability condition (34) and the indirect resonance condition (7) as they differ only by the definitions of $\omega_k$ and the factor $1/\Gamma_k^2$ in the r.h.s. of Eq. (34). To see how this observation allows relating the conditions Eq. (7) and Eq. (10), let us now focus on the eigenstates but from the perspective of the exact equation (24). Substituting Eq. (26) into Eq. (24), we get

$$|E\rangle = \langle \varepsilon | E \rangle \sum_{n=1}^{N} \frac{v_n}{E - E_n} |n\rangle + \langle v | E \rangle \frac{1}{E - \varepsilon} |\varepsilon\rangle, \tag{35}$$

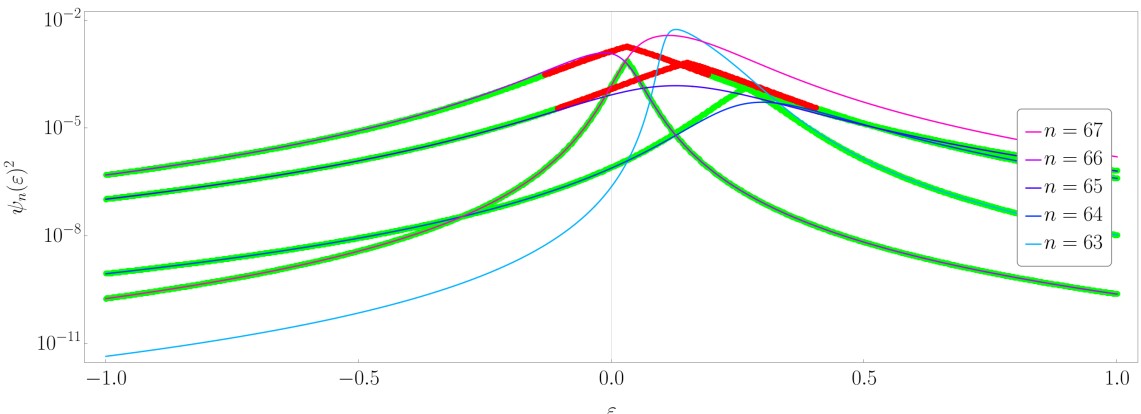

(a) The Gaussian Rosenzweig-Porter model with $N = 2^7$ and $\gamma = 1.5$; the on-site disorder is sampled from the uniform distribution $\varepsilon \in [-1, 1]$.

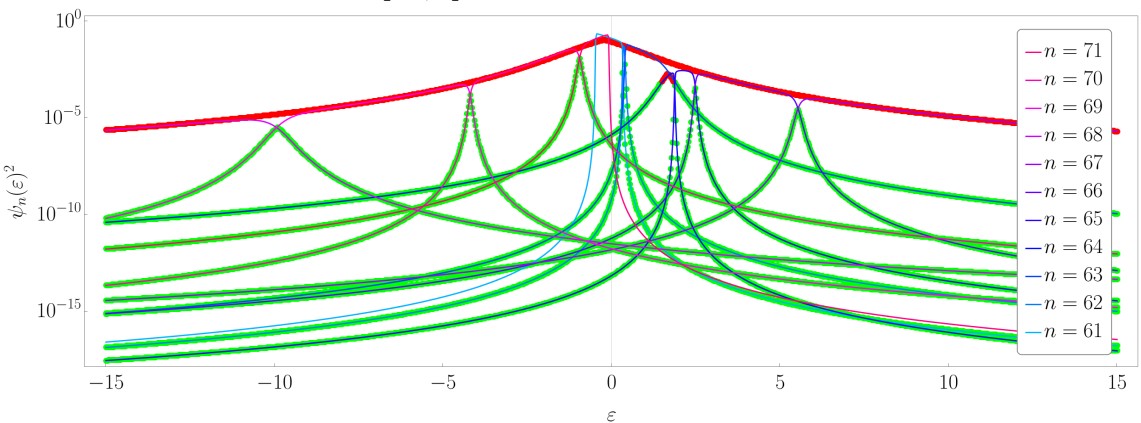

(b) The Bernoulli Rosenzweig-Porter model with $K = 3$, $N = 2^7$; the on-site disorder is sampled from the uniform distribution $\varepsilon \in [-15, 15]$.

Figure 10: The validity check of Eq. (39). Each plot shows a single realization of a random matrix from the corresponding ensemble, with no averaging taken. Different continuous lines show the exact $(N + 1)^{\text{th}}$ site's occupations $\psi(\varepsilon)^2$ as functions of the corresponding onsite energy $\varepsilon$; the legends show the indices of the considered eigenstates. The points show Eq. (39); their color shows if the condition (32) in the form $|\Delta_k| < \min\{|E_k - E_{k\pm1}|\}/2$ holds or not: green means it holds, and red means it does not.

where $|n\rangle$ are the eigenstates of $H_N$ corresponding to the eigenenergies $E_n$. Then, multiplying Eq. (24) by $\langle v|$ to get $\langle v|E\rangle = \langle \varepsilon|E\rangle \, G_{vv}^0(E)$ and using the secular equation Eq. (27) in the form $G_{vv}^0(E) = E - \varepsilon$, we get

$$|E\rangle = \langle \varepsilon|E\rangle \left( |\varepsilon\rangle + \sum_{n=1}^{N} \frac{v_n}{E - E_n} |n\rangle \right), \tag{36}$$

from where, employing the normalization condition $\langle E|E\rangle = 1$, we obtain

$$\psi_E(\varepsilon)^2 = \langle \varepsilon|E\rangle^2 = 1 \bigg/ \left( 1 + \sum_{n=1}^{N} \frac{v_n^2}{(E(\varepsilon) - E_n)^2} \right) = \frac{dE(\varepsilon)}{d\varepsilon}; \tag{37}$$

here, $E(\varepsilon)$ is one of the $N + 1$ branches of the solution of Eq. (27), and the very last equality can be checked by directly differentiating Eq. (27).

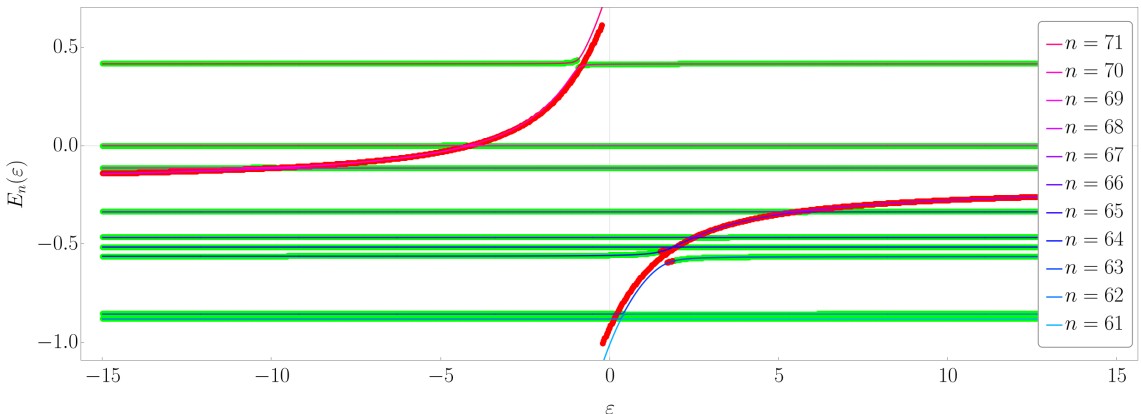

Figure 11: A few branches of $E(\varepsilon)$ corresponding to Fig. 10b; the solid lines represent the exact branches, and the dashed horizontal lines mark the values of $E_n$. The points are plotted according to Eq. (30).

Finally, by passing from $E(\varepsilon)$ to $\Delta_k(\varepsilon)$, isolating the term $v_k^2/\Delta_k^2$ from the rest of the sum, multiplying the nominator and denominator by $v_k^2$ and using Eq. (28), we rewrite the r.h.s. of Eq. (37) in the form

$$\psi_E(\varepsilon)^2 = v_k^2 \Bigg/ \left( \left( E_k - \varepsilon + \Delta_k - \sum_{n \neq k} \frac{v_n^2}{\omega_{kn} + \Delta_k} \right)^2 + v_k^2 \left( 1 + \sum_{n \neq k} \frac{v_n^2}{(\omega_{kn} + \Delta_k)^2} \right) \right), \qquad (38)$$

which is still exact but seems to be a bit more suitable for asymptotic analysis as it reminds the Lorentzian form of the local density of states. To highlight the analogy even more, we can assume Eq. (32) to hold, neglect $\Delta_k$ where needed, and get

$$\psi_E(\varepsilon)^2 \overset{(32)}{\sim} \frac{v_k^2}{(\omega_k + \Delta_k \Gamma_k^2)^2 + v_k^2 \Gamma_k^2}, \qquad (39)$$

or, proceeding further with (34), get

$$\psi_E(\varepsilon)^2 \overset{(34)}{\sim} v_k^2/\omega_k^2. \qquad (40)$$

As we can see, the indirect resonance condition (7) (or, rather, (34)) plays the role of the applicability condition of the eigenstates' perturbation theory expression (8) (or, rather, (40)). The regularized occupation ansatz (9), in its turn, behaves similarly to the Lorentzian approximation (39), so we expect $1/\Gamma_k^2$ to serve as a microscopic analog of the phenomenological threshold $C/\Omega$.

The numerical assessment of the approximation (39) is shown in Fig. 10. Looking at these plots, one may notice a curious fact that could have been seen from the approximation's derivation itself: each fixed-index curve plotted according to Eq. (39) approximates not one but two eigenstates with neighboring eigenenergies! Indeed, the approximation led to Eq. (28) states that the branch $E(\varepsilon)$ corresponding to $\Delta_k(\varepsilon) = E(\varepsilon) - E_k$ should be the closest one to $E_k$, and this non-analytic closeness condition forces our approximation to jump between different branches of $E(\varepsilon)$: for the large negative $\varepsilon$ the closest $E(\varepsilon)$ is larger than $E_k$, while for the large positive $\varepsilon$ the closest $E(\varepsilon)$ is smaller than $E_k$.

One more surprising thing one can notice from the comparison of the exact and approximate occupations in Fig. 10 is that sometimes the points' color turns red, signifying the approximation condition no longer holds, while the approximation still works pretty well. To

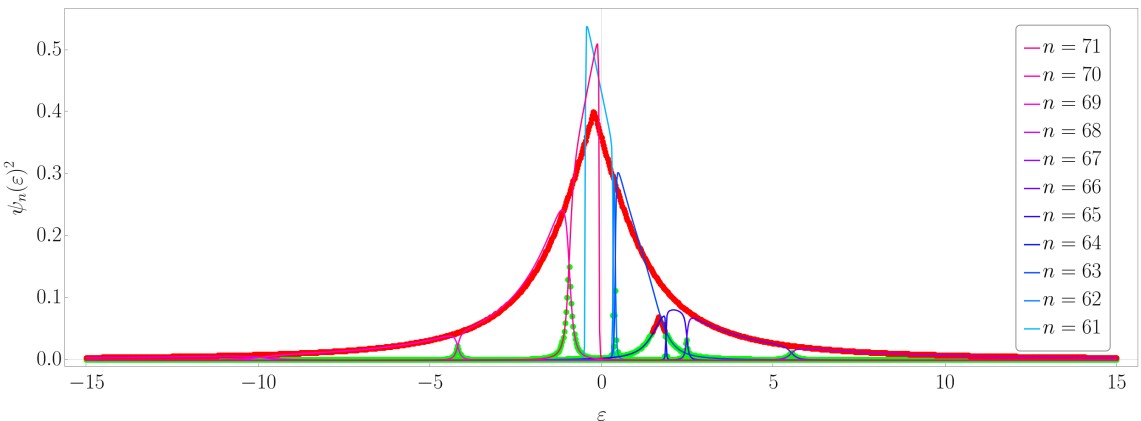

(a) The same eigenstates as in Fig. 10b approximated using Eq. (39) but in a linear scale. One can clearly see that the red envelope describes the occupations almost perfectly despite (32) does not hold.

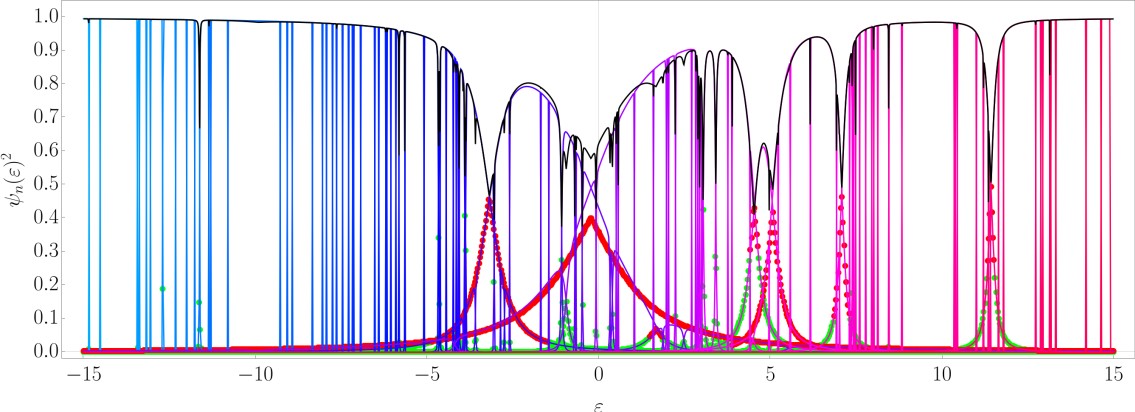

(b) The whole spectrum of the eigenstates for the same realization as in Fig. 10b and Fig. 12a with the additional $(N+1)$th occupation (black solid line) obtained from the normalization condition by subtracting a sum of all other approximate occupations from one.

> Figure 12: An illustration of how well (39) can work even beyond its mathematically justified range of applicability (32). Notice that the Lorentzian approximations (39) to the eigenstates of $H_{N+1}$ are enumerated by the index $k$ which takes only $N$ values corresponding to the eigenstates of $H_N$; hence, there is always one eigenstate which can never be approximated directly by Eq. (39) but, providing (39) works well for all other eigenstates, can be found from the normalization, as shown in the lower panel, Fig. 12b. This eigenstate corresponds to the largest occupation, always counted as a head, and corresponds to the '1+' part of the equations for $\Omega$, e.g., (11).

understand why, consider Fig. 11: due to the spread of the values of $v_n^2$, terms from (27) with relatively large values of $v_n^2$ are directly affecting not only their corresponding level spacings but also some next neighbors' ones. And, while, formally speaking, (39) can never hold for $\Delta_k > \delta$, the rare large realizations of $v_k$, providing $v_n$'s with $n$ close to $k$ are much smaller, force the corresponding asymptotic expressions to "jump" between different branches, effectively describing an envelope of several different wave functions; see Fig. 12 for even more impressive demonstration of this effect. However, the quality of this occasionally good envelope approximation inevitably degrades as $\Delta_k$ grows because $\Gamma_k$ and $\omega_k$ do not contain a valuable dependence on $\Delta_k$, which, eventually, cannot be ignored. In the next section, we derive a correct applicability condition for this "envelope approximation" and show its connection to the self-consistent resonance condition introduced in Sec. 3.

## 6.2 Self-consistent probabilistic approximation

To start with, let us reconsider the transformations leading from Eq. (37) to Eq. (39), and try to understand, without referencing the secular equation (27), why the Lorentzian approximation can work beyond the range of applicability set by Eq. (32). When isolating the term $v_k^2/\Delta_k^2$ in the exact occupation expression (37), we set the stage for separating the contribution of this individual term in the sum $\sum_{n=1}^N v_n^2/(E-E_n)^2$ from the collective contribution of all other terms. This means that the resulting approximation's applicability should be decided by the relation between the collective and individual contributions; hence, the applicability criterion should look like

$$\frac{v_k^2}{\Delta_k^2} \gg 1 + \sum_{n \neq k} \frac{v_n^2}{(\Delta_k + \omega_{kn})^2}, \tag{41}$$

where we did not rely on any approximation for $\Delta_k$ and just used its exact value. Also, we now do not require $\Delta_k$ to be the least-absolute-value solution for a given $k$; instead, we fix a branch of $E(\varepsilon)$ and look at all possible expressions for it, $E(\varepsilon) = E_k + \Delta_k$, $k = 1, ..., N$. Thus, if, for a given $\varepsilon$ and a fixed branch of $E(\varepsilon)$, the condition (41) breaks down for all $k$, the approximation of an individual contribution fails, and we find ourselves inside the head of the wavefunction where the occupation is determined by the collective contribution. This would mean that the r.h.s. of Eq. (41) is of the order of the corresponding inverse occupation $1/\psi_E(\varepsilon)^2$ for any $k$, meaning that removing any single term from the sum does not significantly affect its value. Given that we do not know how to write this collective contribution explicitly, we propose a probabilistic analog of the exact condition (41) in the form

$$\frac{v_k^2}{\Delta_k^2} \gg \Gamma_{head}^2, \tag{42}$$

where we defined $\Gamma_{head}^2$ as a random variable emulating the distribution of the r.h.s. of Eq. (41) when its fluctuations with $k$ are negligible. The corresponding probabilistic version of the Lorentzian occupation approximation (39) is then

$$\psi_E(\varepsilon)^2 \sim \begin{cases} 1/\Gamma_{head}^2, & \Delta_k^2/v_k^2 \gtrsim 1/\Gamma_{head}^2 \\ \Delta_k^2/v_k^2, & \Delta_k^2/v_k^2 \ll 1/\Gamma_{head}^2 \end{cases}. \tag{43}$$

In contrast to all other criteria and approximations discussed in Sec. 6.1, this pair cannot be compared with an individual realization of an eigenstate, but it is designed to predict a *distribution* of the tails' components of the eigenstates.

Let us now discuss how to estimate the distribution of $\Delta_k$. According to the exact expression (37), a site's occupation is equal to the derivative of the corresponding eigenenergy with respect to the site's onsite energy. Hence, $\psi_E(\varepsilon)^2 = d\Delta_k/d\varepsilon$, and we can integrate the r.h.s. of Eq. (43) to get

$$\Delta_k \sim \begin{cases} \varepsilon/\Gamma_{head}^2 + \text{const}, & \Delta_k^2/v_k^2 \gtrsim 1/\Gamma_{head}^2 \\ v_k^2/(\mathcal{E}_k - \varepsilon) \sim v_k^2/\omega_{head}, & \Delta_k^2/v_k^2 \ll 1/\Gamma_{head}^2 \end{cases}, \tag{44}$$

where $\mathcal{E}_k$ is the integration constant. Because the above arguments do not allow an exact calculation of this constant, we introduce $\omega_{head}$ similarly to how we did earlier with $\Gamma_{head}^2$, i.e., as a random variable emulating the actual distribution of $\mathcal{E}_k - \varepsilon$. Provided the width of the distribution of $\mathcal{E}_k$ is small compared to the onsite disorder, one can assume $\omega_{head}$ to be distributed as $\varepsilon - E$, where $E$ marks the energy under consideration.

Finally, recalling that the piecewise form used in Eqs. (43) and (44) (and even in (9)) is just a way to regularize the otherwise singular expressions, on can rewrite the newly derived

expression in a form closely resembling Eq. 39, namely, as

$$\psi_E(\varepsilon)^2 \sim \frac{v_k^2}{\omega_{tail}^2 + v_k^2 \Gamma_{head}^2}.$$ (45)

Associating $1/\Gamma_{head}^2$ with $\psi_{head}^2$ from Eq. (8), we finally obtain the mathematical justification for the extended range of applicability of the Lorentzian approximation Eq. (39) and realize it is just the microscopic version of the self-consistent criterion phenomenologically introduced in Sec. 3. In fact, we could have used this Lorentzian regularization instead of Eq. (9) already there, but, given that it does not drastically improve predictions while significantly complicates formulas, we prefer the piecewise regularization as given by Eq. (9).

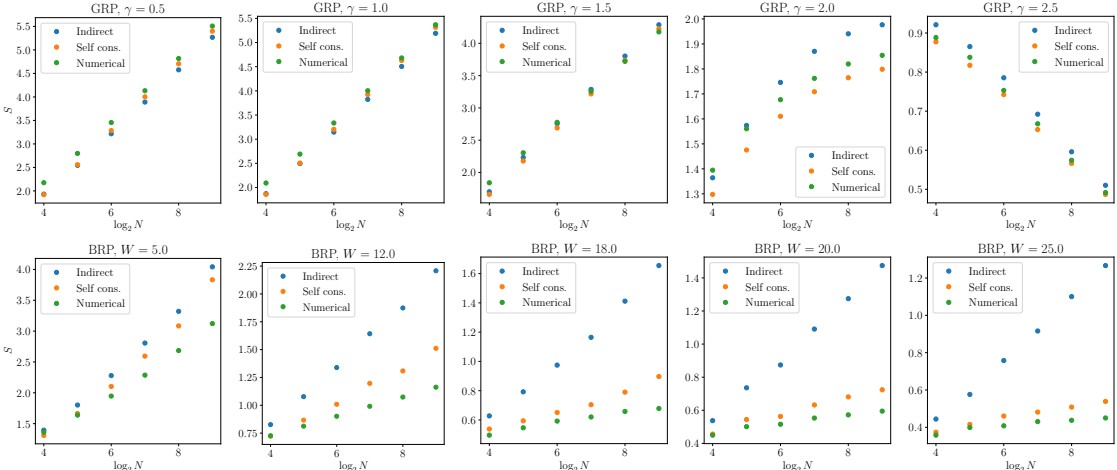

Figure 13: Comparison between different resonance conditions for the Gaussian RP model at different $\gamma$'s (top raw) and the Bernoulli RP model at different $w$'s (bottom row). The blue points (named 'Indirect') are computed using the indirect resonance condition $\Delta_k > \min(|E_{k-1} - E_k|, |E_k - E_{k+1}|)$ with $\Delta_k$ from Eq. (30) to separate tails from heads and using the Lorentzian approximation (39) to calculate $C$ and $S$ via Eq. (11) and Eq. (15). The orange points (named 'Self cons.') are obtained using the self-consistent criterion (10) and the equations below but with $\omega_k$ calculated via Eq. (31). The green points are obtained numerically using the exact eigenfunctions.

As one final remark, let us get back to the threshold problem and its solution we discussed in Sec. 3. As we mentioned there, provided the distribution of the dressed hopping has a characteristic scale, the self-consistent resonance condition (10) is roughly equivalent to the indirect resonance condition (7) derived from the prescription $\Delta < C\delta$, with $C$ being the correct threshold. However, as $C$ is the total weight of the eigenstate's head, it cannot be larger than one. How is it then possible to claim that the self-consistent condition (10) can explain the extended range of applicability of Eq. (39) when the corresponding $\Delta_k$ clearly exceeds the mean level spacing? The answer lies in the absence of the characteristic scale of the Bernoulli RP's dressed hopping distribution. Indeed, since the distribution is clearly heavy-tailed, the threshold argument from the Sec. 3 is not applicable here, and the resonance condition (10) for the Bernoulli RP model goes beyond the condition (7), which we can see in the Fig. 13.

A summary of all the resonance conditions, eigenstate approximations, their applicability conditions, and their interrelations studied throughout the paper, is given in Table 1.

| Phenomenological condition | Its microscopic analog | Wave function profile |
|---|---|---|
| Direct, $\Delta \gtrsim \omega$, (1) | – | – |
| Indirect, $\Delta \gtrsim \min\{\delta, v\}$, (7) | Applicability condition (34) | Singular, (8) & (40) |
| | Applicability condition (32) | Lorentzian, (39) |
| Self-consistent, (10) | Probabilistic condition (42) | Regularized, (9) & (45) |

Table 1: Resonance conditions studied throughout the paper and their relations to the applicability conditions from Sec. 6.1 and to each other.

# 7 Conclusion

In this paper, we have systematically addressed the concept of resonances, intending to bridge the gap between the naive, physically intuitive definition and a predictive tool able to reliably compute relevant quantities such as participation entropies and their corresponding fractal dimensions. We have achieved this goal by introducing a self-consistent resonance criterion, that has many advantages. First of all, it is physically grounded and formally justified, both in terms of a perturbation theory expansion for the wave function of a new site added to the system and via a controlled approximation of the exact size-increment equation describing the site addition (see Sec. 6.1). Moreover, it is free from a problem that is typical of other definitions of resonances: it does not make use of an arbitrary threshold to decide whether a site is in resonance or not, but the self-consistency automatically amends this issue.

We have also proposed an ansatz for approximating the wave function, which is tightly bonded to the resonances picture and distinguishes between components according to the resonance criterion prescription: the support set components are approximated with Haar random vectors, while the tails are approximated according to the second-order perturbation theory. Within this ansatz, we could predict analytically the participation entropy and the support set dimension of the finite size Gaussian Rosenzweig-Porter model, in perfect agreement with the numerical results and with other approaches (see Ref. [47] and App. B). We could also make new predictions for the $\beta$-function of the model. The analytical solution of this model has been possible because of the known distribution of the dressed hoppings. We have also tested our method on other, more complicated random matrix models, for which the distribution of the dressed hoppings is not known, forcing us to compute it numerically. Also in those cases, we have shown how our method captures correctly the behavior of the system, with the analytical predictions that seem to approach the numerical curves as the size grows. However, for these other models, the (generic) ansatz we have proposed for the wave function's head's distribution is not as suitable as it was for the Gaussian Rosenzweig-Porter model, thus leading to a discrepancy in the numerical values; we believe this discrepancy can be reduced by choosing a better, system-specific ansatz for the head's distribution. We leave for future work the goal of finding a more refined ansatz for the ergodic part of the wave function and the analytical computation of the dressed hopping distribution.

Finally, the careful finite-size analysis we performed for the log-normal Rosenzweig-Porter model raised questions about if it can actually serve as a proxy to the Anderson model on RRG, and to what extent. As an alternative, we introduced the Bernoulli Rosenzweig-Porter model which is expected to serve as a better proxy while preserving the simplicity of the RP models and thus saving the hope of obtaining its analytical description, sooner or later.

## Acknowledgements

The authors are grateful to Boris Altshuler, Vladimir Kravtsov, and Antonello Scardicchio for useful discussions and feedback, and collaboration on related topics and to Federico Balducci and David Long for careful reading of the manuscript. The authors also thank Anushya Chandran and David Long for useful discussions and for pointing out that the self-consistent resonance condition solves the threshold problem.

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

# A  Exact entropy for the rotational-invariant random matrices

In rotational-invariant random matrix ensembles such as the Gaussian Orthogonal Ensemble (GOE), Gaussian Unitary Ensemble (GUE), or Gaussian Symplectic Ensemble (GSE), the eigenvectors, due to the rotational symmetry, are distributed uniformly over all directions, meaning that the individual components' distribution does not depend on the basis we are working on. An example of a basis-independent vector distribution is a multivariate normal distribution with a unit covariance matrix. However, the multivariate normal distribution does not respect the normalization; hence, we take the *normalized* multivariate normal distribution, e.g., the corresponding occupations $|\phi(i)|^2$ of the site $i$ can be described by the expression

$$|\phi(i)|^2 = \frac{\sum_{\alpha=1}^{\beta} x_\alpha^2(i)}{\sum_{\alpha=1}^{\beta} x_\alpha^2(i) + \sum_{j\neq i}^{N} \sum_{\alpha=1}^{\beta} x_\alpha^2(j)}, \tag{A.1}$$

where $N$ is the size of the matrix, $x_\alpha(i)$ are the i.i.d. standard Gaussian random variables, and $\beta$ is the Dyson index ($\beta_{GOE} = 1$, $\beta_{GUE} = 2$, $\beta_{GSE} = 4$). Thus, the distribution of $|\phi(i)|^2$ can then be written as

$$p_{\phi^2}(v) = \int_0^\infty \mathrm{d}x \mathrm{d}r\, \delta\left(v - \frac{x}{x+r}\right) \chi_\beta^2(x) \chi_{\beta(N-1)}^2(r), \tag{A.2}$$

where $\chi_k^2(x)$ stands for the PDFs of the chi-squared distribution with $k$ degrees of freedom. After taking this integral, one finds that $p_{\phi^2}(v) \propto v^{\beta/2-1}(1-v)^{\beta(N-1)/2-1}$; i.e., $|\phi(i)|^2$ is distributed according to the beta distribution, $|\phi(i)|^2 \sim \mathcal{B}(\beta/2, \beta(N-1)/2)$. So, having the explicit exact expression for the PDF of the occupations, we can obtain the exact expression for the corresponding participation entropy as

$$S_\beta(N) = -N\left\langle |\phi(i)|^2 \ln |\phi(i)|^2 \right\rangle = H(\beta N/2) - H(\beta/2), \tag{A.3}$$

where $H(x)$ is the Harmonic number.

Due to their maximal ergodicity, the eigenstates of the rotational-invariant ensembles can serve as a reasonable model for the heads of the more complicated ensembles' eigenstates. For example, for $\beta = 1$, the total entropy of such a head according to the ansatz Eq. (9) would be

$$S_{head}(\Omega, C) = \Omega s_{head} = -\Omega\left\langle C\phi^2 \ln\left(C\phi^2\right)\right\rangle = CS_1(\Omega) - C\ln C. \tag{A.4}$$

# B  Another analytical approach to the Gaussian RP model

In the right panel of Fig. 3, we compare our analytical prediction for the support set dimension beta function of the Gaussian RP model with the exact numerical results and with the analytical results based on Ref. [47]. In this section, we summarize the idea of that paper and describe how we apply it to our case.

The main idea of Ref. [47] lies in the ansatz for the distribution of the Gaussian RP eigenfunctions' components which is composed of two parts: the Lorentzian local density of states ('a Breit-Wigner formula with the spreading width $\Gamma$ calculated by the Fermi golden rule') and the Gaussian fluctuations ('a local Porter-Thomas law') on top of it. The distribution is then reads as

$$p_{\psi_E}(x) = \int \frac{\rho(\varepsilon)\mathrm{d}\varepsilon}{\sqrt{2\pi \langle|\psi_E(\varepsilon)|^2\rangle}} \exp\left\{-\frac{x^2}{2\langle|\psi_E(\varepsilon)|^2\rangle}\right\}, \quad \langle|\psi_E(\varepsilon)|^2\rangle \sim \frac{\widetilde{C}}{(E-\varepsilon)^2 + \Gamma(E)^2}, \tag{B.1}$$

where $\widetilde{C}$ is a constant to find from the normalization, $\rho(\varepsilon)$ represents a PDF of the onsite energies, and $\Gamma(E) \sim \pi N^{1-\gamma} \rho(E)$ providing $\gamma > 1$ and $N \gg 1$. This ansatz has its problems: e.g., due to the infinite support of the Gaussian, it always gives a non-zero probability for the normalization to be violated. But, for large enough $N$ and $\gamma < 2$,[7] the corresponding effects should be negligible, and this is what the authors of Ref. [47] prove with their beautiful numerics using $\rho(\varepsilon) \propto e^{-\varepsilon^2/2}$. So, let us now use this ansatz to calculate the participation entropy $S(N)$ in the middle of the spectrum of the Gaussian RP model with the box-distributed onsite energies.

First, let us compute the normalization constant $\widetilde{C}$ using $\rho(\varepsilon) = 1/2w$ for $-w < \varepsilon < w$ and $\rho(\varepsilon) = 0$ otherwise. From the requirement $\langle \psi_0^2 \rangle = 1/N$ where we explicitly put $E$ to zero, we find

$$
\frac{1}{N} = \int_{-\infty}^{\infty} x^2 p_{\psi_0}(x) \mathrm{d}x = \int_{-w}^{w} \frac{\mathrm{d}\varepsilon}{2w} \frac{\widetilde{C}}{\varepsilon^2 + \Gamma^2} = \widetilde{C} \frac{\tan^{-1}\left(\frac{w}{\Gamma}\right)}{w\Gamma} \quad \Longrightarrow \quad \widetilde{C} = \frac{w\Gamma}{N \tan^{-1}\left(\frac{w}{\Gamma}\right)}, \quad \text{(B.2)}
$$

with $\Gamma = \Gamma(0) = \pi N^{1-\gamma}/2w$; the relation between $\widetilde{C}$ and $C$ from the main text will be discussed later. Next, we compute the participation entropy as

$$
\begin{aligned}
S &= -N \int_{-\infty}^{\infty} x^2 \ln(x^2) p_{\psi_0}(x) \mathrm{d}x \\
&= N \int_{-w}^{w} \frac{\mathrm{d}\varepsilon}{2w} \int_{-\infty}^{\infty} \frac{-x^2 \ln(x^2) \mathrm{d}\varepsilon}{\sqrt{2\pi \langle |\psi_E(\varepsilon)|^2 \rangle}} \exp\left\{ -\frac{x^2}{2 \langle |\psi_E(\varepsilon)|^2 \rangle} \right\} \\
&= N \int_{-w}^{w} \frac{\mathrm{d}\varepsilon}{2w} \langle |\psi_0(\varepsilon)|^2 \rangle (\gamma + \ln(2) - 2 - \ln(\langle |\psi_0(\varepsilon)|^2 \rangle)) \\
&= \gamma + \ln(2/\widetilde{C}) - 2 + N\widetilde{C} \int_{-w}^{w} \frac{\mathrm{d}\varepsilon}{2w} \frac{\ln(\varepsilon^2 + \Gamma^2)}{\varepsilon^2 + \Gamma^2},
\end{aligned} \quad \text{(B.3)}
$$

where $\gamma$ stands for the Euler gamma. The last integral can be expressed using a generalized hypergeometric function (or a polylogarithm), and the result for the corresponding support set dimension's $\beta$-function can be seen as the dashed lines in the right panel of Fig. 3. As follows from the comparison, the result is equivalent to ours for large $N$ but deviates from the numerical results and the self-consistent resonance counting prediction for intermediate sizes as well as at the Anderson transition, $\gamma = 2$. A reason for this discrepancy may lie in the nature of the Breit-Wigner approximation as it assumes the broadening $\Gamma$ to self-average, while this assumption fails at intermediate sizes, critical points, and localized phases.

Finally, let us link the $\gamma$-dependent quantity $\widetilde{C}$ to the head's weight $C$ from the main text, which appears to be $\gamma$-independent, see Fig. 4. This apparent discrepancy originates from the difference in the definitions of the quantities; and, since the definition of the head's weight $C$, in contrast to the one of $\widetilde{C}$, includes the hopping distribution explicitly, there can be no exact relation between these two quantities. However, one can try to estimate $C$ from Eq. B.1 as a weight of the head of the averaged wave function defining the 'head' as anything larger than a half-maximum of $\langle |\psi_0(\varepsilon)|^2 \rangle$, i.e., as

$$
C \approx N \int_{|\varepsilon| < \min\{w, \Gamma\}} \frac{\mathrm{d}\varepsilon}{2w} \frac{\widetilde{C}}{\varepsilon^2 + \Gamma^2} = \frac{N\widetilde{C}}{w\Gamma} \tan^{-1}\left( \frac{\min\{\Gamma, w\}}{\Gamma} \right) = \frac{\tan^{-1}(\min\{1, w/\Gamma\})}{\tan^{-1}(w/\Gamma)}. \quad \text{(B.4)}
$$

In the large-$N$ limit, $\Gamma \propto N^{1-\gamma}$, and for $1 < \gamma < 2$ (for $\gamma > 2$, (B.1) systematically violates normalization), our estimate gives $C \approx 1/2$, while for $\gamma < 1$, it is $C \approx 1$, which is in fact exactly what we see in Fig. 4!

---

[7]For $\gamma > 2$, the problem with the normalization becomes essential, leading to incorrect results evident in Fig. 3.