# Peer review of "Investigating finite-size effects in random matrices by counting resonances"

_SciPost Physics_

## Round 2 · Referee Report · Anonymous (Referee 1) · 2024-8-4

Strengths

  1. well written
  2. detailed derivation of results
  3. Comparison of numerics on different models supports their phenomenological theory

Report

In the manuscript, the authors present a self-consistent approach to resonance counting in disordered quantum systems. They propose a resonance criterion that connects this to a phenomenological ansatz for the wave functions, allowing for the calculation of quantities such as participation entropy, support set dimension, and related beta function. They provide further microscopic support for their phenomenological theory and test its predictions against exact diagonalization results for various Rosenzweig-Porter (RP) models.

The manuscript reports on original and interesting work, providing insights into resonance counting and finite-size scaling of related quantities in disordered quantum systems. I also find the manuscript well written and, therefore, support its publication after the authors address the following questions and comments.

-A curious result in my opinion is the scaling property of the weight of the wave functions head. The authors show that in the thermodynamic limit the latter approaches unity in the localized and ergodic phases, implying that all weight is in the head and tails do not contribute. While this is intuitive, I find it somehow surprising that the weight takes a single fixed value in the entire intermediate fractal regime (rather than a \gamma-dependent value). Do they have any explanation for (i) why it’s a single value in the entire fractal regime, and (ii) why this is 1/2? This should be related to the change of wave function statistics from Porter Thomas distribution to the modified distribution, discussed in Appendix B? Can they comment more on this?

-I have two basic questions regarding a statement made about the resonance counting based on dressed hopping elements.
The authors comment that the estimate for the Gaussian RP model “severely underestimates the support set volume (…)”. This also applies for conventional RMT Hamiltonian (where a similar estimate N^{1/2} also severely underestimating the number of resonances)? They continue “(…) unable to correctly locate the ergodic fractal transition”. I think some more explanation of this point would be helpful, since the change in scaling from usual RMT can be identified with the ergodic fractal transition? (In the end, the Thouless inspired criterium gives the “square” of the condition derived from dressed resonances).

-In criterium (7), the authors carefully distinguish contributions from direct and indirect resonances, but in criterium Eq. (10) such distinction does not play a role? Can they comment on this?

-I would find it helpful to add a brief discussion on which of the considered quantities are basis independent and which not.

-If possible, a more detailed discussion on the deviations between analytical predictions and exact diagonalization for the log-normal and Bernoulli RP models would be beneficial. For example, they briefly mention a relation to matrix sparsity, could they provide further commentary on this?

-A more general question (out of curiosity and not the focus of their work) is which of their findings they expect to transfer to “true” disordered many-body systems with long range interactions. These systems share similarities with the Gaussian RP model but are more sparse and exhibit correlated disorder. How would this affect their results?

Recommendation

Publish (meets expectations and criteria for this Journal)

---

## Round 2 · Referee Report · Anonymous (Referee 2) · 2024-8-27

Strengths

See report

Weaknesses

See report

Report

Despite a long history and a very large research effort, Anderson localization remains a difficult subject. The reason is that even though the propagation of waves in a random medium, a problem that includes single-body quantum evolution in a random potential, is qualitatively well-understood, it is hard to make precise and sharp statements, and a lot of the understanding is based on heuristic ideas and arguments.

Resonances between sites with nearby orbital energies is a transport mechanism concept that has played a central role in the studies of localization for a long time, but it is difficult to assign a sharp meaning to this concept. The goal of the present paper is to use this idea to study the energy levels and eigenfunctions in a random medium, in particular as a tool to capture finite size effect.

The main result is the self-consistent resonance condition, a mean-field like approximation, that is then applied to several random-matrix models of localization. Special attention is given to the Gaussian Rosenzweig-Porter model, where the theory approximates quite closely the exact statistics; this is unsurprising, given that the GRP is a mean-field model of localization. Even though the predictive power of the theory is smaller for the other random matrix models, qualitative behavior in them is captured by the theory, in particular the behavior in the thermodynamic limit. It is of interest to examine the validity of the theory and its possible improvements for finite-dimensional lattices, but this question is clearly outside the scope of the current paper.

While not a breakthrough, the resonance condition proposed in this work is a significant step in the effort to better understand quantum energy levels and wavefunctions in random environments, so I recommend publication of the paper.

Requested changes

  1. The paper assigns particular importance to the log-normal Rosenzweig-Porter model, but the discussion of the analysis of this model at the top of page 14 is hard to follow, and in particular it is not clear which parts of the discussion are based on the self-consistent resonance condition and which parts on numerical diagonalization. This is also a problem in Fig 7 which presents results for this model, where in particular the relation between the two panels is not clarified.

  2. Can the authors provide some insight regarding the source of the discrepancy between the self-consistent resonance theory and exact results for the LNRP and Bernoulli RP models, and how one may improve the accuracy of the approximation?

  3. Section 6 on the microscopic approach, while clearly related to the other sections, seems to stand apart in terms of ideas and methods. It would be better to either more strongly tie it to the rest of the paper or relegate it to another publication.

  4. While the graphs themselves are clear and polished, the captions are not always self-contained, sometimes not very clearly labeled, and color choices are sometimes not ideal. Note a mistake in the legends of Fig 6

  5. The sharp transition from head to tail statistics in the wavefunction ansatz (as depicted in Fig 2) seems too crude. Could it be a significant source of error? The authors address this question in section 6, concluding that replacing it by a smoothed transition does not improve the approximation. Nevertheless, this a possible weakness of the theory which merits further scrutiny.

Recommendation

Publish (easily meets expectations and criteria for this Journal; among top 50%)

---

## Editorial Decision

resubmitted